# When Much Is Too Much—Compared to Light Exercisers, Heavy Exercisers Report More Mental Health Issues and Stress, but Less Sleep Complaints

**DOI:** 10.3390/healthcare9101289

**Published:** 2021-09-28

**Authors:** Sanobar Golshani, Ali Najafpour, Seyed Sepehr Hashemian, Nasser Goudarzi, Fatemeh Shahmari, Sanam Golshani, Masthaneh Babaei, Kimia Firoozabadi, Kenneth M. Dürsteler, Annette Beatrix Brühl, Jalal Shakeri, Serge Brand, Dena Sadeghi-Bahmani

**Affiliations:** 1Kermanshah University of Medical Sciences, Kermanshah 6714869914, Iran; snb.gln@gmail.com (S.G.); fatemesh9373@gmail.com (F.S.); jshakerimd@yahoo.com (J.S.); 2Student Research Committee, Faculty of Medicine, Kermanshah University of Medical Sciences, Kermanshah 6714869914, Iran; ali98aaa@yahoo.com; 3Department of Psychology, Allameh Tabataba’i University, Tehran 1489684511, Iran; sepehr.hashemian@gmail.com; 4Department of Psychiatry, AJA University of Medical Sciences, Tehran 1411718541, Iran; nassergoodarzi@yahoo.com; 5Department of Cardiology, AJA General Hospital, Kermanshah 6714869914, Iran; Sanamgolshani7@gmail.com; 6School of Medicine, Urmia University of Medical Sciences, Urmia 5714783734, Iran; mastanebabai91@gmail.com (M.B.); kimia.firoozabadi65@gmail.com (K.F.); 7Psychiatric Clinics, Division of Substance Use Disorders, University of Basel, 4002 Basel, Switzerland; Kenneth.Duersteler@upk.ch; 8Center for Addictive Disorders, Department of Psychiatry, Psychotherapy and Psychosomatics, Psychiatric Hospital, University of Zurich, 8001 Zurich, Switzerland; 9Center for Affective-, Stress- and Sleep Disorders (ZASS), Psychiatric Clinics (UPK), University of Basel, 4002 Basel, Switzerland; Annette.Bruehl@upk.ch (A.B.B.); dena.sadeghibahmani@upk.ch (D.S.-B.); 10Sleep Disorders Research Center, Kermanshah University of Medical Sciences, Kermanshah 6714869914, Iran; 11Substance Abuse Prevention Research Center, Kermanshah University of Medical Sciences, Kermanshah 6714869914, Iran; 12Department of Sport, Exercise, and Health, Division of Sport Science and Psychosocial Health, University of Basel, 4052 Basel, Switzerland; 13Department of Psychiatry, School of Medicine, Tehran University of Medical Sciences, Tehran 1417466191, Iran; 14Department of Psychology, Stanford University, Stanford, CA 94305, USA

**Keywords:** heavy and light exercisers, mental toughness, stress perception, sleep quality, general health

## Abstract

Background: Physical inactivity has become a global somatic and mental health issue. To counterbalance, promoting regular physical activity appears plausible, above all among adults, where physical inactivity is particularly high. However, some, but sparse, research also indicates that excessive exercising might be associated with unfavorable mental health dimensions. Here, we tested the hypothesis that excessive exercising was associated with more mental health issues. To this end, we assessed mental health issues, stress, mental toughness, and sleep disturbances among heavy and light adult exercisers. Methods: A total of 200 adults (mean age: 35 years; 62% females) took part in the study. Of those, 100 were heavy exercisers (18–22 h/week), and 100 were light exercisers (1–6 h/week). Participants completed questionnaires covering sociodemographic information, mental health issues, perceived stress, mental toughness, and sleep disturbances. Results: Compared with light exercisers, heavy exercisers reported higher mental health issues, more stress, but also higher mental toughness scores and less sleep disturbances. Higher age, lower mental toughness scores, heavy exerciser-status, and more sleep disturbances predicted higher mental health complaints. Conclusions: Compared with light exercising, heavy exercising might be associated with more mental health issues. As such, it appears that the association between exercise frequency, intensity, and duration and psychological well-being might be related to an optimum point, but not to a maximum point. In a similar vein, heavily exercising athletes, their coaches, parents, and representatives of sports associations should get sensitized to possible adverse psychological effects of excessive physical activity patterns.

## 1. Introduction

Approximately 31% of the global population aged ≥15 years engages in insufficient physical activity, and such kind of low activity is known to contribute to the death of approximately 3.2 million people every year [1,2,3,4]. To illustrate: Adult people in the USA spend 55% of their waking time (7.7 h a day) engaged in sedentary behaviors; people in European countries spend 40% of their leisure time (2.7 h a day) watching television [5]. An increasing pattern of sedentary lifestyle and insufficient physical activity (IPA) were associated with poor psychological functioning among adolescents aged 11–17 years [6] and adults aged 18 to 80 years [7]. Korean adults reported 8.3 h of sedentary time a day [8]. Importantly, among Koreans aged 50 years and older, longer sitting time and lower physical activity indices were associated with chronic low back pain [9] and the risk of cardiovascular diseases at long-term [10]. Therefore, both reducing sedentary behaviors and increasing physical activity appeared to be important factors to promote public health [11]. Not surprising, mental health issues [12] such as stress [13] and sleeping problems [14] were tightly associated with low physical activity patterns [15,16]. We took these observations into account and investigated the associations between mental health issues, stress, mental toughness, and sleep among adult heavy and light exercisers.

In a very similar vein, Iran is among the countries with medium to high insufficient IPA indices in global reports: In 2011, national reports revealed an IPA rate of 39.1% among the adult population [17]. Eight years later, the prevalence of IPA raised to 54.7%, with a considerable difference between the two genders (males: 45.3%, females: 61.9%) [18]. In this view, physical inactivity is particularly challenging among adults in their thirties, as this period of life is a psychosexually, psychosocially, and economically demanding developmental stage of adulthood; as such, physical inactivity appears to affect the economically most active population [19] and the population with the highest pressure to prevail and assert for successful mating [20,21,22]. We took this observation into consideration and assessed mental health issues, stress, and sleep disturbances among Iranian adult heavy and light exercisers in this age range.

While there is extant literature to show that physical inactivity is related to higher mental health issues (see above), there is also extant literature to show the favorable association between regular physical activity patterns and lower mental health complaints (e.g., [23]). Such favorable associations have been observed among children [24,25,26,27,28], adolescents [6,29,30,31,32,33,34,35], and adults [8,13,36,37,38,39,40,41,42,43,44,45,46,47,48,49,50], and among clinical samples, such as individuals with major depressive disorder [32,51,52,53,54,55,56,57,58,59,60,61,62,63,64,65,66,67,68] or multiple sclerosis [69,70,71,72,73,74,75,76,77,78,79,80,81,82,83]. And such observations hold also true as regards the associations between higher physical activity patterns and better coping with stress [60,84,85,86,87,88,89,90]. More generally, higher physical activity patterns were associated with more restoring sleep, lower perceived stress levels, and higher mental toughness scores. As such, the results mentioned above suggest that “more (physical activity) leads to more (psychological well-being)”.

However, such claims have been challenged: Sparse, inconsistent, but increasing data show that excessive exercising might be associated with symptoms of mental health issues [91,92,93,94,95,96,97,98]. Similarly, at least among adolescents and longitudinally, physical activity patterns were completely unrelated to participants’ mental health [99], challenging the claim that, almost by nature, higher physical activity patterns lead more favorable mental health. In this regard, we also note that 12–14% of promising junior elite athletes reported symptoms of burnout [100,101].

Thus, recent research points out to a dilemma: While physical inactivity is related to more physical and mental health issues, this might be associated with excessive exercising, too. Typically, psychological dimensions related to physical activity patterns are mental health, mental toughness, perceived stress, and sleep quality. Given this background and given that, to our knowledge, data from Iran on this topic are missing so far, we assessed mental health, stress, mental toughness, and sleep disturbances as a proxy for psychological functioning among heavy exercisers (3 h/day; 6 days/week), compared with light exercisers (1–6 h/week).

Almost by definition, a personality trait tightly related to sports performance is mental toughness. Indeed, there is sufficient evidence of a favorable association between higher physical activity patterns and higher mental toughness scores [34,101,102,103,104,105,106,107,108,109,110,111,112,113,114,115,116,117,118]. Here, mental toughness is understood as the personality trait conferring the ability to control one’s life and emotions, to perceive demands and expectations as a challenge (as opposed to a threat), to stay committed to one’s own aims and goals, and to have confidence both in one’s abilities and in stable relationships. This is labelled the “Four C model” [119]. Importantly, in the area of sports performance, mental toughness is associated with persistence, with being better prepared to deal with challenges, and with finding a way to succeed despite setbacks [110,120,121,122]. In this line, mental toughness favorably moderated the relation between physical activity intention and physical activity behavior. Further, it appears that the origins of mental toughness go back to pre-school age [123]. However, we also note that a recent systematic review showed that despite the widely used term of mental toughness as a personality trait important for performance in the field of sport, no uniform and coherent definition of mental toughness appears to be available [124].

Thus, while the general setup is that higher mental toughness scores are associated with better psychological functioning and higher physical activity performance, on the flip side, there are also some critical statements: Specifically, athletes with high mental toughness scores might be at risk of overtraining [125] and of displaying also negative personality traits, such as the so-called dark trait [126,127,128]. Similarly, athletic identity was associated with higher symptoms of depression, mental toughness, over-adherence to physical activity intensity, playing through pain, and injury severity [129]. We took these conflicting results into consideration and asked if heavy exercisers reported higher mental toughness scores, compared with light exercisers.

Lastly, an important factor for both physical and mental health is restoring sleep. Again, there is sufficient evidence on the associations between restoring sleep and higher physical activity patterns: Interventional [130,131] and associative studies [33,132,133,134,135,136,137] suggested that higher physical activity levels favorably impacted on sleep quality; though, bi-directional and reciprocal processes appeared to be most plausible [138,139]. We considered these observations and asked if heavy exercisers reported less sleep disturbances, compared with light exercisers.

Overall, there is sufficient evidence that physical inactivity is causally related to a broad variety of non-communicable diseases among children, adolescents, and adults. In contrast, regular physical activity levels are associated with a broad range of mental health indices, such as lower risk of psychiatric issues, along with lower stress, better sleep, and higher mental toughness scores as a proxy for high psychological functioning. On the flip side, there is also some, also inconsistent, evidence that heavy exercising might be associated with mental health issues, as excessive mental toughness scores might also lead to adverse health effects. Further, to our knowledge, research on these topics has not been conducted so far among Iranian adults.

As a consequence, the aims of the present study were to address these issues, and given these backgrounds, the following two hypotheses and two research questions were formulated:

First, based on previous research [33,132,133,134,135,136,137,138,139,140], we assumed that compared with light exercisers, heavy exercisers would report less sleep complaints.

Second, following others [60,84,85,86,87,88,89,90], we assumed that compared with light exercisers, heavy exercisers would report lower scores of perceived stress.

Next, the first research question asked was if and to what extent heavy exercisers would report more favorable or more unfavorable mental health scores and mental toughness scores, always compared with light exercisers. This research question is based on more recent research on the adverse effects of excessive exercising on mental health [91,92,93,94,95,96,97,98].

The second research question asked was if sleep, stress, mental toughness, and exercising status could predict mental health issues. To this end, we assessed mental health, perceived stress, mental toughness, and sleep complaints of heavy and light adult Iranian exercisers.

We hold that the results might be of clinical and practical importance: If it turns out that heavy exercising is also associated with adverse health issues, then exercisers, coaches, parents, and representatives of sports associations should get sensitized to possible adverse psychological disadvantages of heavy exercising.

## 2. Methods

### 2.1. Procedure

Adults attending regular physical activities were approached and asked to participate in the present study on physical activity patterns and psychological functioning. Participants were fully informed about the aims of the study and the secure and anonymized data handling. Thereafter, participants signed a written informed consent and completed a series of self-rating questionnaires on mental health, perceived stress, and sleep (see details below). The ethical committee of Kermanshah University of Medical Science, Kermanshah, Iran (Ethics ID: IR. KUMS. REC. 1398. 583) approved the study, which was performed in accordance with the seventh and current [141] edition of the Declaration of Helsinki.

### 2.2. Participants

We selected adults with heavy and light exercising patterns. The procedure was as follows: By referring to the Kermanshah Sports Federation (Kermanshah, Iran), people who exercised heavily and people who exercised lightly, in all sports, such as running, tennis, martial arts, swimming, aerobics, football, basketball, volleyball, and handball, were selected with the same age and, if possible, the same gender.

Inclusion criteria for heavy exercisers were: (1) age between 18 and 65 years; (2) for the last two years, exercising at least six days a week, for at least 3 h a day, both as self-reported and as ascertained by professional coached from the Iranian Sports Federation; (3) compliance with the study conditions; (4) signed written informed consent. Exclusion criteria were: (1) psychiatric issues; to this end, trained psychiatrists and clinical psychologists performed face-to-face clinical interviews [142] based on the DSM-5 [143]; (2) regular intake of mood-altering, sleep-altering, or performance-enhancing substances, such as psychoactive medications or illicit drugs (e.g., steroids, amphetamines, anabolic substances); (3) severe physical issues, such as injuries, chronic inflammations, autoimmune diseases, musculoskeletal issues, chronic pain and similar, as assessed via a brief medical interview by trained medical doctors.

Inclusion and exclusion criteria for light exercisers were identical to the inclusion and exclusion criteria of heavy exercisers, with the following difference: exercising between one to three days a week, for one to two hours a day as self-reported and as ascertained by professional coaches from the Iranian Sports Federation.

### 2.3. Measures

#### 2.3.1. Sociodemographic Information

Participants reported their age (years), gender (male, female), civil status (single; married), current job position (employed; unemployed), and highest educational level (high school diploma; higher educational level).

#### 2.3.2. General Health Questionnaire

To assess mental health issues, the Farsi version [144] of the General Health Questionnaire-28 [145] was employed. The Farsi version had psychometrically satisfying indices (Cronbach’s alphas for somatization, anxiety, social functioning, depression, and the overall score: 0.70, 0.84, 0.93, 0.92, 0.85, respectively). Typical items are somatic symptoms (items 1–7), anxiety/insomnia (items 8–14), social dysfunction (items 15–21), or severe depression (items 22–28). Typical items are: “*Have you recently been feeling perfectly well and in good health?*”; “*Have you recently lost much sleep over worry?*”; “*Have you recently been managing to keep yourself busy and occupied?*”; “*Have you recently felt constantly under strain? *”. Answers were given on 4-point Likert scales ranging from 0 (not at all) to 3 (much more than usual). The following sub-scores were calculated: somatization, anxiety, social function, depression, and the total score. Higher sum scores reflect more impaired or more distressed general health (Cronbach’s alpha: 0.89).

#### 2.3.3. Mental Toughness Questionnaire

The Farsi version [115,116,126,146] of the Mental Toughness Questionnaire [119,147] was employed. The Farsi version had psychometrically satisfying indices (lowest Cronbach’s alpha = 0.80). Here, we used the shortened questionnaire with 18 items, which provides a global mental toughness score. Typical items are: “Challenges usually bring out the best in me”; “I generally feel in control”; “Even when under considerable pressure I usually remain calm”. Answers on the MTQ18 were given on 5-point Likert-type scales ranging from 1 = “strongly disagree” to 5 = “strongly agree.” Items were summed to obtain the overall score, with higher scores reflecting a higher mental toughness (Cronbach’s alpha: 0.89).

#### 2.3.4. Perceived Stress

To assess perceived stress, the Farsi version [148,149] of the Perceived Stress Scale [150] was employed. The Farsi versions had psychometrically satisfying indices (lowest Cronbach’s alpha = 0.72). Typical items are: “In the last month, how often have you been upset because of something that happened unexpectedly?”; “In the last month, how often have you been angered because of things that were outside of your control?”; In the last month, how often have you been able to control irritations in your life?” (reversed scaling). Answers were given on 5-points Likert scales ranging from 1 (never) to 5 (almost always), with higher scores reflecting higher perceived stress (Cronbach’s alpha: 0.89).

#### 2.3.5. Sleep Disturbances

To assess sleep disturbances, the Farsi version [151,152,153,154,155] of the Pittsburgh Sleep Quality Index [156] was employed. The Farsi versions had psychometrically satisfying indices (lowest Cronbach’s alpha = 0.77). As mentioned elsewhere [151], the questionnaire consists of 18 items, which are summed up to the following dimensions: (1) subjective sleep quality; (2) sleep latency; (3) sleep duration; (4) habitual sleep efficiency; (5) sleep disturbances; (6) sleeping medication; (7) daytime dysfunction. Each component score has a possible range of 0 to 3, with higher scores indicating poorer sleep. Further, the PSQI global score of sleep quality ranges from 0 to 21. A sum score of 5 points or higher reflects sleep disturbances (Cronbach’s alpha: 0.81).

### 2.4. Statistical Analysis

First, the inspection with a series of Kolmogorov–Smirnov tests showed that outcome variables were normally distributed.

With a *t*-test and a series of χ^2^-tests, we compared sociodemographic information between heavy and light exercisers.

With a series of Pearson’s correlations, associations between age, general health, perceived stress, mental toughness, and sleep complaints were calculated for the whole sample and also separately for heavy and light exercisers.

A series of *t*-tests was performed to compare means of perceived stress, mental toughness, and sleep complaints between heavy and light exercisers.

A series of multivariate ANOVAs was performed with the factor exerciser status (light; heavy) and dimensions of general health (somatization, anxiety, social function, depression, general health score) as dependent variables.

To predict general health, a multiple regression analysis was performed. Preliminary conditions to perform multiple regression analyses were met [157,158,159]: N = 200 > 100; predictors explained the dependent variables (R = 0.683, R^2^ = 0.466); the number of predictors: 8; 10 × 8 = 80 < N (200), and the Durbin–Watson coefficient was 1.83, indicating that the residuals of the predictors were independent. Furthermore, the variances inflation factors (VIF) were between 1.03 and 1.18, while there are no strict cut-off points to report the risk of multicollinearity, VIF < 1 and VIF > 10 indicate multicollinearity [158,159].

To test for possible interaction effects within the multiple regression model, we followed Aiken and West [160], who proposed to multiply the residuals of the independent factors and to model this product as a function of the categories of one factor. Specifically, we multiplied the residuals of the factors exercise status and sleep disturbances; next, the product was entered into the multiple regression model as a further factor. Lastly, to understand the nature and direction of the interaction, a scatter plot was created with the dimensions exercise status and general health, moderated by the category of sleep disturbances (no vs. yes).

Effect sizes were reported as follows: For *t*-tests, Cohen’s ds were calculated, with the following cut-off values: trivial (ds: 0–0.19), small (ds: 0.20–0.49), medium (ds: 0.50–0.79), or large (ds: 0.80 and greater). For F-tests, effect sizes were reported as partial eta-squared [*η_p_*^2^]) and interpreted as follows: trivial (T) 0.019 < *η_p_*^2^; small (S) = 0.020 ≤ *η_p_*^2^ ≤ 0.059, medium (M) = 0.06 ≤ *η_p_*^2^ ≤ 0.139, or large (L) = *η_p_*^2^ ≥ 0.14 [161,162].

The level of significance was set at alpha < 0.05. All statistical procedures were performed with the SPSS^®^ 25.0 (IBM Corporation, Armonk, NY, USA) for Apple Mac^®^.

## 3. Results

### 3.1. General Information

Table 1 provides the descriptive and inferential statistical overview of sociodemographic information on heavy and light exercisers.

As shown in Table 1, no descriptively or statistically significant differences of sociodemographic data were observed between heavy and light exercisers.

### 3.2. Descriptive Statistics and Correlations between Age, General Health, Perceived Stress, Mental Toughness, and Sleep Disturbances for the Whole Group and Separately for Heavy and Light Exercisers

Table 2 provides the descriptive statistics and correlations between age, general health, perceived stress, mental toughness, and sleep disturbances for the whole group and separately for heavy and light exercisers.

As regards the whole sample, a higher age was associated with higher general health complaints, lower mental toughness scores, and higher sleep disturbances, while age was not associated with perceived stress. Further, higher general health complaints were associated with higher sleep disturbances, while general health was not associated with perceived stress and mental toughness. Higher mental toughness scores were associated with lower sleep complaints.

As regards heavy exercisers, a higher age was associated with higher general health complaints, lower mental toughness, and higher sleep complaints, while age was not associated with perceived stress. Higher general health complaints were associated with higher perceived stress, lower mental toughness, and higher sleep complaints. Higher perceived stress was associated with lower mental toughness and with more sleep complaints. Higher mental toughness scores were associated with lower sleep complaints.

As regards light exercisers, age was not associated with general health, perceived stress, mental toughness, or sleep complaints. Higher general health complaints were associated with higher perceived stress and higher sleep complaints, but not with mental toughness. Higher perceived stress was associated with lower mental toughness, but not with sleep complaints. Mental toughness was unrelated to sleep complaints.

### 3.3. General Health, Perceived Stress, Mental Toughness, and Sleep Complaints; Differences between Heavy and Light Exercisers

Table 3 provides the descriptive and inferential statistical indices for perceived stress, mental toughness, and sleep complaints between heavy and light exercisers.

Compared with light exercisers, heavy exercisers reported higher perceived stress and higher mental toughness (significant *p*-values; always medium effect sizes). Sleep complaints did not differ.

Further, compared with light exercisers, heavy exercisers reported higher somatization, anxiety, depression, and overall scores (significant *p*-values; small effect sizes) and lower social functioning scores (significant *p*-value; small effect size).

### 3.4. Predicting General Health

To predict general health, a multiple regression was performed with the general health score as a dependent variable and perceived stress, mental toughness, sleep complaints, and exerciser status (heavy vs. light)**,** age, gender, and the exercise status-sleep-disturbances interaction as predictors. Table 4 provides all statistical indices.

More impaired sleep, lower mental toughness scores, heavy exercise status, older age, and a higher exercise-status–sleep-disturbance interaction predicted more impaired general health. Perceived stress and gender were excluded from the equation, as these dimensions did not reach statistical significance.

The exercise-status–sleep-disturbances interaction showed that heavy exercisers with more sleep disturbances also reported more psychological health issues.

## 4. Discussion

The key findings of the present study were that compared with light exercisers, heavy exercisers reported more mental health issues, including higher perceived stress, but also higher mental toughness scores and less sleep disturbances. Higher sleep disturbances, lower mental toughness scores and heavy exerciser status predicted higher mental health issues. Further, the combination of being a heavy exerciser with high sleep disturbances predicted higher mental health issues (see Table 4). In our opinion, the present results expand upon the current literature in three important ways:First, among Iranian heavy exercisers, self-reported mental health issues along with higher perceived stress could be observed.Second, such health issues were highly associated with poor sleep.Third, higher mental toughness indices were associated with less mental health issues, but more so among heavy exercisers; in contrast, higher mental toughness scores were associated with lower perceived stress indices among light exercisers.

Overall, the data suggest that among Iranian adults identified as heavy exercisers, mental health issues could be observed. As a consequence, exercisers, coaches, and representatives of sports organizations should be sensitized to the rule of thumb, that “more is not always better”.

Two hypotheses and two research questions were formulated, and each of these is considered now in turn.

In the first hypothesis, we assumed that compared with light exercisers, heavy exercisers reported less sleep disturbances, and data did confirm this. Thus, the present data confirmed previous results [33,132,133,134,135,136,137,138,139,140]. However, we expanded upon previous results in that such associations were observed among heavy adult exercisers in Iran. Next, we also note that other studies also showed exactly the opposite: compared with non-athletes, professional athletes self-reported more sleep disturbances [163,164,165,166].

In the second hypothesis, we predicted that compared with light exercisers, heavy exercisers would report lower stress scores; though, data did confirm exactly the opposite. Given this, the present findings do not match what has been observed elsewhere [60,84,85,86,87,88,89,90]. Results are discussed in more details below, when dealing with the two research questions.

The first research question we asked was if and to what extent heavy exercisers would report more favorable or more unfavorable general health scores and mental toughness scores compared with light exercisers. The answer was not straightforward; while heavy exercisers reported higher mental toughness scores, they also reported more mental health issues. While the former result could be expected from what we know from previous studies [34,101,102,103,104,105,106,107,108,109,110,111,112,113,114,115,116,117,118], the latter result appears to confirm the raising research that excessive exercising could also be associated with mental health issues [91,92,93,94,95,96,97,98]. Thus, we claim that the present data expand upon the sparse but increasing literature on the unfavorable association between excessive exercising and mental health issues, in that such a pattern was also observed among heavily exercising Iranian adults.

The associations between higher mental toughness scores and higher mental health issues among heavy exercisers demand more attention. First, we note that the correlation coefficient was negative (r = −0.29; see Table 2); this would imply that higher mental toughness and lower mental health issues were associated. However, as noted above, heavy exercisers did report both higher mental toughness and higher mental health issues, which appears to be contradictory. To solve this apparent contradiction, we note that athletes with high mental toughness scores might be at risk of overtraining [125] and of displaying also negative personality traits, such as the so-called dark trait [126,127,128]. Similarly, heavy exercisers also reported higher symptoms of depression, mental toughness, over-adherence to physical activity intensity, playing through pain, and injury severity [129]. Given this, we claim that such unfavorable associations could also appear in the present sample of heavy exercisers. Note that results are discussed in more details below when dealing with the two research questions.

In the second research question, we asked, with factors could predict higher mental health issues, and the answer was: poor sleep, lower mental toughness scores, heavy exerciser status, higher age, and an unfavorable exercise-status–sleep-disturbances interaction (see Table 4). Thus, the regression equation confirmed that the combination of several factors predicted higher mental health issues, and again, heavy exerciser status was among these predictors, while perceived stress or gender did not reach statistical significance.

Heavy exerciser status and higher mental health issues were associated, and this pattern of results demands more attention. To this end, we give a closer examination to the exercise-status–sleep-disturbance interaction.

In regard to the exercise-status–sleep-disturbance interaction, we noted that the combination of heavy exercise status and higher sleep disturbances predicted higher general health issues. This pattern of results appears particularly intriguing, as it does not follow the mainstream of interventional [130,131] and associative studies [33,132,133,134,135,136,137], which suggested that higher physical activity levels favorably impacted on sleep quality. Thus, while we asked if heavy exercisers reported less sleep disturbances, compared with light exercisers, the answer was no (see Table 3); however, the combination of being a heavy exerciser and reporting sleep disturbances was important, as this combination impacted negatively on mental health.

Further, the following admittedly speculative assumptions are made:

First, it is conceivable that the extensive exercising duration, frequency, and intensity put the exerciser into a dilemma as regards balancing the budget of time and resources against the time and resources needed for working and family life. To illustrate, among 292 undergraduate students, extreme exercise schedules interfered with their social, occupational, and family lives [167]. Similarly, among professional body builders, excessive exercising increased the risk of issues with social and occupational functions, along with subjective distress, and adverse health consequences [168].

Second, it is conceivable that excessive exercising behavior might be associated with dimensions of (pre-existing) psychopathology, such as maladaptive perfectionism [169] or dysfunctional coping with stress [170].

Third, blending the first and second claim, it is conceivable that excessive exercising was not the trigger of stress, but a dysfunctional answer to stress.

Fourth, physiological dimensions are conceivable; more specifically, at the mitochondrial level, “there is too much of a good thing”; excessive training impaired mitochondrial activity and decreased glucose tolerance, at least among adult healthy volunteers [171,172]. In contrast, among male adults with symptoms of burnout, an exercise-related increased mitochondrial activity over time was associated with a more favorable recovery [53].

To conclude, several hypothetical pathways are conceivable to describe why heavy exercising might have been associated with higher mental health issues.

Despite the novelty of the results, the following limitations warrant against their overgeneralization. First, the cross-sectional design precluded any causality. However, to run multiple regressions, predictors and dependent variables must be defined a priori. In this view, we assumed that poor sleep, low mental toughness scores, and heavy exerciser status predicted general heath; however, it is also conceivable that poor mental health instigated the increase of exercise frequency, intensity, and duration as a strategy to cope with psychological health issues. Meanwhile, there is sufficient evidence that moderate to vigorous physical activity patterns favorably impact on psychological well-being [173]. Furthermore, perceived stress negatively impacted on physical activity and exercise [174]. Similarly, a bidirectional interactional effect between physical activity patterns and psychological well-being might be the most appropriate model. Second, sample selections were such to clearly identify participants’ physical activity duration and frequency based on both participants’ self-reports and their supervisors’ information. However, a detailed assessment of participants’ physical activity patterns would have allowed to introduce this dimension as a further factor. Third, unlike other case-control studies, we did not run an a priori sample size calculation. While the lack of sample size calculations bears the risk to miss small but important mean differences, which could have yielded further important patterns of results, on the flip side, we relied on effect sizes, which by nature are not sensitive to sample sizes. Fourth, higher physical activity traits are associated with lower [170] or higher self-esteem [175,176], along with a higher risk of eating-disordered behavior [170]; it is therefore conceivable that self-esteem, eating-disordered behavior, and further latent and un-assessed psychological dimensions might have biased two or more variables in the same or opposite directions. Fifth, by definition, a longitudinal design would have allowed to gain more insight into causal associations between excessive exercising and mental health issues.

## 5. Conclusions

Compared with light exercising, heavy exercising was associated with mental health issues and higher perceived stress, but also with higher mental toughness scores and lower sleep complaints. Overall, it appeared that “more (heavy exercising) is not better (mental health)”; as such, heavy exercisers, but also coaches, parents of promising junior athletes, and representatives of sports associations should consider paying special attention to mental health issues.

## Figures and Tables

**Table 1 healthcare-09-01289-t001:** Overview of sociodemographic information on heavy and light exercisers.

		Exercisers		Statistics
Dimensions		Heavy	Light	
N		100	100	
		M (SD)	M (SD)	
Age (years)		35.45 (13.48)	34.17 (12.30)	*t* (198) = 0.04
		N (%)	N (%)	
Gender	Male	(34) 34	(42) 42	χ^2^ (N = 200; df = 1) = 1.36, *p* = 0.24
	Female	(66) 66	(58) 58	
Civil status	Single	(49) 49	(51) 51	χ^2^ (N = 200; df = 1) = 0.80, *p* = 0.78
	Married	(51) 49	(49) 49	
Employment	Employed	(47) 47	(50) 50	χ^2^ (N = 200; df = 1) = 0.18, *p* = 0.67
	Unemployed	(53) 53	(50) 50	
Educational level	High school diploma	53 (27)	55 (28)	χ^2^ (N = 200; df = 1) = 0.19, *p* = 0.86
	Higher education	47 (23)	45 (22)	

**Table 2 healthcare-09-01289-t002:** Descriptive statistics and Pearson correlation coefficients between age, general health, perceived stress, mental toughness, and sleep disturbances for the whole group and separately for heavy and light exercisers.

	Age			GHQ			PSS			MT			PSQI		
Dimensions	T	HE	LE	T	HE	LE	T	HE	LE	T	HE	LE	T	HE	LE
Age	-	-	-	0.31 **	0.50 ***	0.05	0.02	−0.13	0.14	−0.26 **	−0.36 ***	−0.17	0.25 **	0.54 ***	−0.08
GHQ	-	-	-	-	-	-	−0.03	0.43 ***	0.23	−0.09	−0.29 **	0.12	0.64 ***	0.67 ***	0.69 ***
PSS	-	-	-	-	-	-	-	-	-	−0.26 **	−0.17 *	−0.24 **	−0.09	0.22 **	0.06
MT	-	-	-	-	-	-	-	-	-	-	-	-	−0.11	−0.31 ***	0.11
PSQI	-	-	-	-	-	-	-	-	-	-	-	-	-	-	-
M	35.17	35.17	34.35	20.52	21.82	19.21	34.57	36.06	33.09	31.30	32.51	30.09	4.81	4.54	5.07
SD	(13.48)	(12.93)	(13.58)	19.62	12.59	8.05	4.93	4.17	5.20	5.98	5.57	6.16	2.52	2.48	2.54

Notes: * = *p* < 0.05; ** = *p* < 0.01; *** = *p* < 0.001; T = total sample; HE = heavy exercisers; LE = light exercisers; GHQ = general health questionnaire; higher scores reflect a more impaired health; PSS = perceived stress scale; a higher score reflects a higher perceived stress; MT = mental toughness; a higher score reflects a more pronounced mental toughness; PSQI = Pittsburgh sleep quality index; a higher score reflects a more disturbed sleep. M = mean; SD = standard deviation.

**Table 3 healthcare-09-01289-t003:** Inferential statistical indices of general health, perceived stress, mental toughness, and sleep complaints between heavy and light exercisers.

	Exercisers		Statistics	
	Heavy	Light	*t*-Tests	Effect Sizes
N	100	100		
	M (SD)	M (SD)		Cohen’s d
Perceived stress	36.06 (4.17)	33.08 (5.20)	*t* (198) = 4.47 ***	0.776 [M]
Mental toughness	32.51 (5.57)	30.09 (6.16)	*t* (198) = 2.91 **	0.054 [M]
Sleep complaints	5.07 (2.55)	4.54 (2.47)	*t* (198) = 1.49	0.237 [S]
General health			F-tests (MANOVA)	Partial eta-squared
Somatization	6.09 (3.50)	5.14 (2.67)	F (1, 198) = 4.64 *	0.023 [S]
Anxiety	6.10 (3.19)	4.92 (2.75)	F (1, 198) = 7.83 **	0.038 [S]
Social functioning	5.66 (3.56)	6.93 (2.24)	F (1, 198) = 9.04 **	0.044 [S]
Depression	3.97 (3.66)	2.22 (2.67	F (1, 198) = 14.89 ***	0.070 [M]
Total score	21.82 (12.59)	19.21 (8.05)	F (1, 198) = 3.05 *	0.089 [M]

Notes: MANOVA = multivariate analysis of variance; * = *p* < 0.05; ** = *p* < 0.01; *** = *p* < 0.001; S = small effect size; M = medium effect size.

**Table 4 healthcare-09-01289-t004:** Multiple linear regression with general health as outcome variable, and perceived stress, mental toughness, sleep disturbances, exerciser status, age, and gender as predictors.

Dimension	Variables	Coefficient	Standard Error	Coefficient β	*t*	*p*	R	R^2^	Durbin–Watson
General Health	Intercept	9.607	2.33	–	4.116	0.000	0.683	0.466	1.68
	PSQI	2.601	0.229	0.617	11.310	0.000			
	Mental toughness	−0.221	0.114	−0.134	−2.175	0.03			
	Exercise status ^1^	−3.989	1.15	−0.118	−3.580	0.000			
	Age	0.120	0.043	0.158	2.93	0.004			
	Exercise status × sleep complaints interaction ^2^	−0.932	0.460	−0.110	−2.030	0.044			
	Excluded variables: Perceived stress; gender; *ts* < 1.0; *ps* > 0.30								

Notes: PSQI = Pittsburgh sleep quality index; higher scores reflect more impaired sleep; ^1^ = heavy exercisers; ^2^ = light exercisers.

## Data Availability

Data are made available upon request to experts in the field and upon through explanations of why and how data are used.

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
