# Peer review of "When Much Is Too Much—Compared to Light Exercisers, Heavy Exercisers Report More Mental Health Issues and Stress, but Less Sleep Complaints"

_healthcare, 2021, doi:10.3390/healthcare9101289_

Round 1
Reviewer 1 Report
Exercise intensity is compared to determine if some mental health positive or negative may rise in heavy exercisers against mild exercisers. The outcomes are surprising but of interest.
The authors found a link between heavy exercise and mental health issues.
Stress coping was also negatively correlated with heavy exercise.
I believe that the results are really of great interest and would like the authors to further discuss their findings. I might have missed this in the manuscript, nevertheless, I’d like to ask the authors the following :
Can you please add some considerations on the stress originating from other factors than exercise a bit more?
e.g.: Can you add some figures and statistics on the kind of occupational stress experienced by the two groups, it may be that the heavy exercisers are already more stressed by some work or family issues and thus they “choose” heavy exercise to evacuate stress. But by adding more stress during sports they finally a negative outcome. I believe that this is an important issue to be discussed and your data may answer in an efficient way. Of course the same question can be considered for light exercisers.
Thanks in advance for your consideration.
Author Response
We thank Reviewer # 1 vfor the valuable comments and suggestions, which helped us to improve the quality of the manuscript. Please find the revision (with track changes) and the detailed point-by-point-response attached as separate files.

Reviewer 2 Report
GENERAL COMMENTS
The aim of this paper was to examine the mental health issues, stress, mental toughness and sleep disturbances among heavy and light adult exercisers. Although this article addresses an interesting topic, many issues should be addressed before publication.
SPECIFIC COMMENTS
INTRODUCTION
The introduction needs major revision and clarification. First, the aim of the study is not clear. The relationship from mental health issues, mental toughness, stress and sleep disturbance is not well explained. Also, the way the authors build up their introduction does not lead to the research question. Although many of the necessary information regarding the background is already written down, the authors should re-structure their introduction, explaining why their research is important. The authors described some studies separately, but the discussion of all four factors together is very shallow. Thus, it is recommend that the authors expand this part: how it is related to heavy and light exercisers. More importantly, this should lead to a clear objective of the study.
METHODS
The methods section needs major revision. As it stands, it is not possible to replicate their study.
Please provide sample size estimation for the study. Please indicate what is the study design and sampling method, what is your reference population.
Please mention the validity and reliability of the Farsi version questionnaires (all four
questionnaires)
Table 3, is the F-tests from MANOVA, if yes, please indicate in the note.
Please give reference about the medium and small effect size for partial eta-squared. Please provide the mean difference and it 95%CI.
Please provide exact p-value, we do not know what is * and **.
Table 4, did you control for other possible confounder variables such as age, gender? You should include them in the multiple linear regression to see whether they are significant factors to the general health.
I think the author missed a mental toughness review paper?
Liew, G. C., Kuan, G., Chin, N. S., & Hashim, H. A. (2019). Mental toughness in sport. German Journal of Exercise and Sport Research, 1-14. https://doi.org/10.1007/s12662-019-00603-3
DISCUSSION
In the discussion section, the authors should further discuss their findings and the implication of these findings. They should also discuss their findings in more depth. However, in this section, the authors present many new results. These results should be moved to the results section. The limitation is also not well thought. Please revise.
Please check the references, not following Healthcare format.
Thank you.
Author Response
We thank Reviewer #2 for the valuable comments and suggestions, which helped us to improve the quality of the manuscript. Please find the revision (with track changes) and the detailed point-by-point-response attached as separate files.

Reviewer 3 Report
The author has explored if heavy exercisers have more mental health issues than light exercisers among adults by testing 1). Whether sociodemographic information affected the amount of exercise; 2). Correlation among mental health issues; 3). The difference of mental health issues that heavy and light exercisers have; 4). Regression model to assess the relationship between general health and other health issues. The paper has plenty of statistical results, but there are some questions about the methods and results below:
- For 3.2 and Table 2, you used analysis of correlation for mental health issues and age, so what kind of correlation coefficients (Pearson, Spearman, Kendall) was used for the test? Did you do any normality test before the analysis of correlation to choose the correct correlation coefficients? Please provide more details.
- For 3.3 and Table 3 upper parts, you did multiple t-tests for PSS, MT and PSQI to find if there was significant difference between heavy and light exercisers for each issue. But in 3.2 you already found correlations between these issues. As these issues were not independent, do you need to consider about the interaction effects between these issues? For example, heavy exercisers associated with higher perceived stress in 3.3, and from the results in 3.2, higher perceived stress associated with more sleep complaints. So heavy exercisers might have more sleep complaints based on the deduction. This might be different from your result. Please consider about the contradiction of your results.
- For 3.3 and Table 3 lower parts, you did multiple one-way ANOVAs for 4 conditions: somatization, anxiety, social functioning and depression, but still only compared the difference between heavy and light exercisers. As you use T-tests for PSS, MT and PSQI, why did you use ANOVA instead of T-test for general health issues? Please explain.
- For 3.4 and Table 4, you fitted a linear regression model by using PSQI, MT and exercise status to predict the GH. Your GH score was already calculated by : Somatization, anxiety, social function, depression scores. The correlation between GH and other issues has already shown in 3.2, so what is your motivation to use these issues to estimate this score? As you already showed that these issues (PSQI, MT and exercise status) were highly correlated to each other, did you consider about the multicollinearity might lead to poor performance and overfitting for this regression model? The R2 score was 0.46 which was not high enough, so I cannot believe the result that more impaired sleep, lower mental toughness scores and heavy exercise status lead to higher general health. You also need to assess your model by cross validation. Please consider about the feasibility of using three issues to predict the GH score, and take assessment of the model before you generate the result.
Author Response
We thank Reviewer # 3 for the valuable comments and suggestions, which helped us to improve the quality of the manuscript. Please find the revision (with track changes) and the detailed point-by-point-response attached as separate files.

Round 2
Reviewer 2 Report
Unfortunately, the authors got for a quick fix approach, it is incorrect. Please address the comments fully and not just mentioning that "The Farsi version had psychometrically satisfying". It is unacceptable. Please give full Cronbach alpha for each subscale. Next, please give the full ethical committee approval code. I had checked with the ethical committee of Kermanshah University of Medical Sciences, Kermanshah, Iran but could not find the actual proposal for the application.
Author Response
Again, we thank Reviewer #2 for their kind suggestions. Please find the responses attached as a separate file. Thank you again for the care devoted to evaluate the manuscript.

Reviewer 3 Report
The paper has good improvement with some changes in result parts:
- It's clear to see the correlation analysis and some interaction analysis betwee n the factors.
- The table looks better to show the results and statistical results.
- I still think the regression analysis is a little redundant since you already did a lot of correlation and0 ANOVA analysis between these factors. But it's not bad to provide more details about the result.
- If possible, please provide more details about the assessment of the regression analysis, of how you split the train/test datasets, cross validation (K-Fold), and test the R2 score (average) of that result. R2 = 0.46 is not bad if you use the different dataset for training and testing. But if you use the same dataset for training and testing, 0.46 is not enough.
Author Response
Again, we thank Reviewer #3 for their kind suggestions. Please find the responses attached as a separate file. Thank you again for the care devoted to evaluate the manuscript.
